# Polycomb repressive complex 2 structure with inhibitor reveals a mechanism of activation and drug resistance

Alexei Brooun[1,*], Ketan S. Gajiwala[1,*], Ya-Li Deng[1], Wei Liu[1], Ben Bolaños[1], Patrick Bingham[2], You-Ai He[1], Wade Diehl[1], Nicole Grable[1], Pei-Pei Kung[1], Scott Sutton[1], Karen A. Maegley[2], Xiu Yu[1] & Al E. Stewart[1]

Polycomb repressive complex 2 (PRC2) mediates gene silencing through chromatin reorganization by methylation of histone H3 lysine 27 (H3K27). Overexpression of the complex and point mutations in the individual subunits of PRC2 have been shown to contribute to tumorigenesis. Several inhibitors of the PRC2 activity have shown efficacy in EZH2-mutated lymphomas and are currently in clinical development, although the molecular basis of inhibitor recognition remains unknown. Here we report the crystal structures of the inhibitor-bound wild-type and Y641N PRC2. The structures illuminate an important role played by a stretch of 17 residues in the N-terminal region of EZH2, we call the activation loop, in the stimulation of the enzyme activity, inhibitor recognition and the potential development of the mutation-mediated drug resistance. The work presented here provides new avenues for the design and development of next-generation PRC2 inhibitors through establishment of a structure-based drug design platform.

[1] Worldwide Medicinal Chemistry, Worldwide Research and Development, Pfizer Inc., San Diego, California 92121, USA. [2] Oncology Research Unit, Worldwide Research and Development, Pfizer Inc., San Diego, California 92121, USA. * These authors contributed equally to this work. Correspondence and requests for materials should be addressed to A.B. (email: alexei.brooun@pfizer.com) or to K.S.G. (email: ketan.gajiwala@pfizer.com).

PRC2 is composed of four core components: EZH2, EED, SUZ12 and RbAp48, although it may interact with several other proteins[1]. Each catalytic cycle of PRC2 transfers a methyl group from the cofactor S-adenosyl-L-methionine (SAM) to the ε-amino group of H3K27. The trimethylated H3K27 (H3K27Me3), product of PRC2 catalysed reaction, is thought to recruit other factors such as PRC1 resulting in the silencing of genes, some of which are tumour suppressors. PRC1 is a multi-protein complex that ubiquitinates histone H2A at Lys119, and frequently co-occupies target sites in the genome with PRC2 (ref. 2). Recently EED has been implicated in the recruitment of PRC1 to the H3K27Me3 (ref. 3). EED also plays a role in the positive feedback loop by sensing the H3K27 methylation state and modulating the enzyme activity in transmission of H3K27Me3 mark[4,5]. RbAp48 is thought to regulate the substrate specificity of the PRC2 (ref. 6). In addition to the role in activation, a zinc-finger motif outside of the VEFS domain of SUZ12 facilitates PRC2 recognition of the genomic target[7]. Efficient binding to H3K27Me3 as well as the propagation of the trimethyl mark requires all three subunits, EZH2, EED and SUZ12 (ref. 8).

The catalytic machinery of PRC2 resides entirely in the C-terminal SET domain of EZH2, although EZH2 itself is neither stable nor active. Structural analysis of EZH2 catalytic domain (520–746) containing pre-SET and SET domains supports the notion that isolated catalytic domain is inactive and sheds some light on how this inactive conformation is maintained[9,10]. Absence of the cofactor SAM and H3K27 peptide recognition by the isolated catalytic domain further underscores its catalytic incompetence[9]. Minimally, interactions with EED and the VEFS domain of SUZ12 (SUZ12-VEFS) are necessary to stimulate the methyltransferase activity of EZH2 (ref. 11). Low-resolution electron microscopy structure places SUZ12-VEFS in close proximity with EZH2 catalytic domain while EED interacts with N-terminus of EZH2 (ref. 12). This picture is consistent with the recently reported structures of PRC2 from a thermophilic fungus, *Chaetomium thermophilum* (*Ct*)[5]. Gain-of-function mutation or upregulated expression of PRC2 components have been implicated in various cancers[13–18]. Several pyridone series PRC2-selective inhibitors are currently under clinical investigation for the treatment of lymphomas[19–24]. These inhibitors are SAM-competitive, although their binding mode is unknown.

Here we report the inhibitor-bound crystal structures of the wild-type (WT) and the oncogenic Y641N PRC2 consisting of human EED, human SUZ12-VEFS and engineered American chameleon (*Anolis carolinensis*) EZH2 (*Ac*EZH2) subunits. We refer to it as *Hs/Ac*PRC2. For the first time the structures reveal the binding mode of the pyridone-based PRC2 inhibitor, and reconcile the apparent incongruities of the hotspots for EZH2 mutations identified in *in vitro* models of acquired drug resistance. Unexpectedly, part of the inhibitor recognition site is formed by the N-terminal EZH2 activation loop which plays a key role in the activation of SET domain. The interaction of EED and SUZ12-VEFS with the activation loop is required for the formation of catalytically competent PRC2. Hydrogen/deuterium exchange mass spectroscopy (HDX-MS) analysis of the oncogenic mutant PRC2 suggests that Y641N substitution has far-reaching consequences on the EZH2 protein dynamics rather than just creating a more spacious substrate-binding site.

## Results

**Protein engineering of vertebrate PRC2.** PRC2 is a challenging target for structural characterization due to its inherently dynamic structure. We focused our reductionist approach on functional complex containing EZH2, EED and SUZ12-VEFS.

(For the work presented here, PRC2 refers to three-component complex unless mentioned otherwise.) Analytical size exclusion chromatography (aSEC)[25] detected aggregation of a significant fraction of purified human PRC2 (*Hs*PRC2) with unphosphorylated Ser583 in SUZ12-VEFS. Introduction of the phosphomimetic mutation S583D in SUZ12-VEFS resulted in PRC2 with an improved solution monodispersity. Initial crystallization experiments with *Hs*PRC2 were not successful. In our attempt to facilitate crystallization by exploring sequence diversity we screened 27 PRC2 consisting of various EZH2 orthologues co-expressed with *Hs*EED (81–441) and *Hs*SUZ12-VEFS S583D (545–726). PRC2 with *Ac*EZH2 (*Hs/Ac*PRC2) distinguished itself with the increased catalytic activity on oligonucleosomes compared with *Hs*PRC2. HDX-MS has been used previously in structural genomics projects to identify constructs amenable to protein biophysical studies[26]. HDX-MS mapping of PRC2 identified flexible regions 1, 2 and 3, which were systematically 'engineered out' to facilitate crystallization of the complex (Fig. 1). EZH2 variants with deletion of region 2 maintained histone methyltransferase activity upon assembly into PRC2 (see below). Interestingly, in the recently reported *Ct*PRC2 structure EZH2 subunit lacks dynamic region 2 connecting Cys3His motif and SANT2 (ref. 5). Most of the constructs generated as a part of our protein engineering effort had good aSEC profiles and retained pyridone derivative inhibitor recognition. To further differentiate between PRC2 variants, we utilized small-angle X-ray scattering (SAXS) data to identify protein complexes with reduced flexibility. Parameters used for evaluation were the extracted radius of gyration ($R_g$), mass[27], volume[27], Porod exponent[28] and maximum dimension[29]. The Porod exponent provides an estimate of the flexibility within a particle with a value of 4, consistent with a compact particle, while a value of 2 is indicative of an unfolded polypeptide. The measured values for Hs/AcPRC2 (Δ 329–415 EZH2) showed promising characteristics (lowest $R_g$, volume, apparent mass and highest Porod exponent) relative to others (Supplementary Table 1).

While we were able to crystallize the *Hs/Ac*PRC2 (Δ 329–415 EZH2), the low-resolution anisotropic diffraction (4–7 Å) did not allow us to solve the structure. Finally, the dynamic linker containing basic nuclear localization site preceding EZH2 catalytic domain identified by HDX-MS was replaced with flexible Gly-based linker. The resulting three-component complex was amenable to crystallization, allowing establishment of the first known structure-based drug design platform for PRC2.

**Biochemical characterization of PRC2.** *Ac*PRC2, *Hs/Ac*PRC2 and *Hs/Ac*PRC2_X produced in this work have robust methyltransferase activity on rH3 protein and H3 peptide (1–35), which is comparable to the intact *Hs*PRC2 four-protein complex (*Hs*PRC2_4; Fig. 2a,b). The low-methyltransferase activity of all PRC2 variants on rH3K27me2 is consistent with the known preference of *Hs*PRC2_4 for catalysing first methylation reaction (Fig. 2b). At the same time we have observed discrepancy in nucleosomal methyltransferase acitivity for the same set of PRC2 complexes (Fig. 2a). Previous studies of *Drosophila melanogaster* PRC2 (*Dm*PRC2) containing EZH2, EED and SUZ12-VEFS showed that it retains enzymatic activity on both oligonucleosomes and H3/H4 tetramers. *Hs*PRC2 produced in our work has identical subunit composition but contains N-terminally truncated EED (81–441). N-terminus of EED is thought to contain putative H3-binding site, and its deletion could result in low-methyltransferase activity of *Hs*PRC2 on oligonucleosomes[30] (Fig. 2a). Concurrently, we identified *Hs/Ac*PRC2 with measurable methyltransferase activity on oligonucleosomes (Fig. 2a). Removal of dynamic region 2 (Fig. 1b) in *Ac*EZH2 resulted in the loss of methyltransferase activity of the crystallized

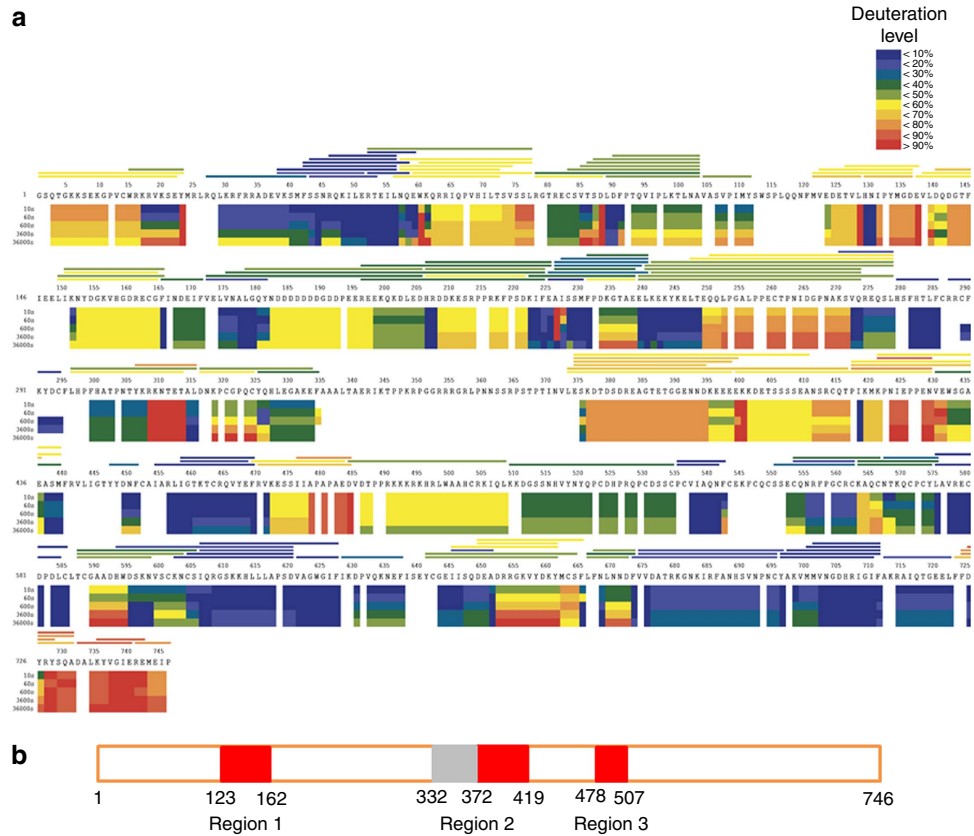

**Figure 1 | Dynamics of HsEZH2 in the context of five-protein HsPRC2.** (**a**) HDX-MS 'heat map' represents the solution dynamics of full-length HsEZH2: red regions indicate protein exposed to solvent and readily available for backbone exchange. Blue regions are blocked from exchange because they are: (1) buried in protein (2) involved in hydrogen bonding (secondary structure) or (3) blocked by protein–protein interactions. The horizontal bars at the top of the heat map represent average deuteration for each of the 147 individual peptides (90% sequence coverage) monitored during the experiments performed in triplicate. (**b**) HDX-derived dynamic 'hot regions' (red) of HsEZH2. Homologous dynamic and 'no coverage' (grey) regions in both HsEZH2 and AcEZH2 were explored with internal deletions.

complex (Hs/Ac PRC2_X) on the oligonucleosomes which could be rescued by replacement of EED (81–441) with full-length EED in PRC2_X_FL_EED (Fig. 2a).

**Overall PRC2 structure.** Human and AcEZH2 share 95% sequence identity (Supplementary Fig. 1). Numbering of human EZH2 sequence is used here. The structurally characterized Hs/AcPRC2 retains methyltransferase activity on both H3 peptide (1–35) and rH3 (Fig. 2a,b), maintains preference for catalysing the first methylation reaction (Fig. 2b)[31] and exhibits similar sensitivity to inhibitor 1 as intact HsPRC2_4 (Table 1, Supplementary Fig. 2; Fig. 3a).

Structures of CtPRC2 consisting of EZH2—SUZ12-VEFS fusion and EED subunits have recently been reported in basal and stimulated states[5]. CtEZH2 and AcEZH2 sequences are 38% identical within the SET domain and divergent elsewhere. Similarly Ct and HsEED sequences share 30% sequence identity with substantial insertions and deletions, while orthologous SUZ12-VEFS share <20% sequence identity. Despite overall structural similarities with CtPRC2 structure, there are significant differences between CtPRC2 and Hs/AcPRC2. The vertebrate PRC2 structure, reported here for the first time, offers insights into the interplay of PRC2 activation, inhibitor recognition and development of the mutation-mediated drug resistance.

The subunit architecture of AcEZH2 (Fig. 3) includes a long N-terminal helix that spans residues 11–62, followed by a β-hairpin between residues 83–96 and largely extended structure up to residue 124. CtEZH2 shows a shorter N-terminal helix that

is terminated at the equivalent of position 48 of the AcEZH2, followed by the β-hairpin that is 10 residues longer than in AcEZH2 (Supplementary Fig. 3a,b). The open architecture of the N-terminal region is almost entirely sustained by its intimate interactions with EED in both PRC2 systems (Fig. 3a, Supplementary Fig. 3a). Within the N-terminal segment, the stretch of residues 108–124 seemingly plays an important role in the activation of the SET domain. We refer to it as the activation loop. The equivalent stretch of residues in CtEZH2, referred to as the SET-activating loop[5], is conserved in length and structure, but is divergent in the amino acid sequence. Residues 125–165 are not modelled due to the lack of electron density. In this aspect, the PRC2 structure reported here resembles the basal state of CtPRC2. Dubbed stimulation-responsive motif, this region is ordered in the CtPRC2 structure in the presence of H3K27Me3 peptide.

Residues 166–246 define the SANT1, first of two SANT domains with three helices. Distribution of acidic residues (Fig. 4a) and the calculated isoelectric point (pI) of 5.2 is consistent with its expected role in the recognition of basic histone substrate[32]. The most apparent difference between HsEZH2 and AcEZH2 sequences (Supplementary Fig. 1) is located in SANT1: a segment consisting of nine Asp residues in HsEZH2, which is evolutionarily lost in AcEZH2 (as also in CtEZH2). SANT1 packs intimately against significantly basic N-terminal helix (Fig. 4a). The equivalent SANT1L region of CtEZH2 is substantially elongated due to the sequence insertion relative to AcEZH2. As a result it shows a different structure (Supplementary Fig. 3b), although still packing

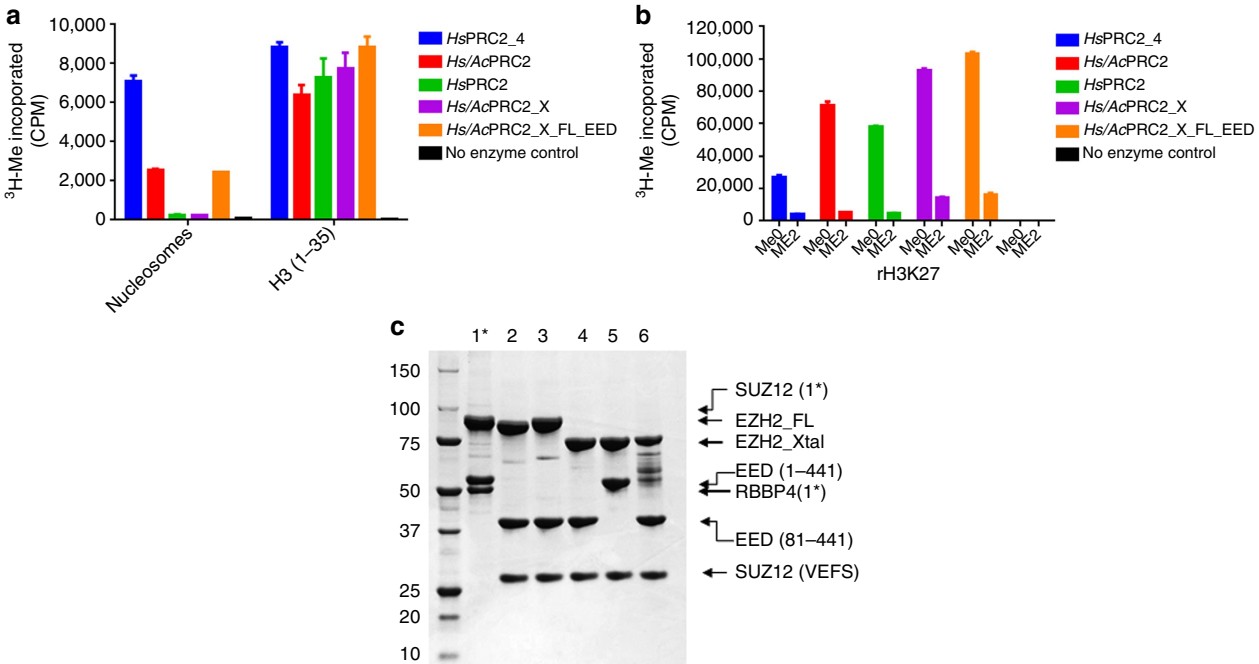

**Figure 2 | Biochemical characterization of PRC2.** (**a**) PRC2 methyltransferase activity on in-house nucleosomes (see the 'Methods' section) and H3 peptide (1–35; Anaspec); *Hs/Ac*PRC2_X is the crystallized PRC2 and *Hs/Ac*PRC2_X_FL_EED is with full-length EED. (**b**) PRC2 methyltransferase activity on rH3Me0 (NEB) and rH3Me2 (Active Motif). Total counts at 30 min from three separate experiments are plotted as mean ± s.d. (**c**) SDS-PAGE analysis of PRC2 complexes used for biochemical analysis: lane 1—intact *Hs*PRC2_4 (full-length EZH2 (EZH2_FL) and SUZ12 co-migrate on SDS-PAGE due to similar molecular weight; additional band below EED is RBBP4), lane 2—*Ac*PRC2, lane 3—*Hs*PRC2, lane 4—*Hs/Ac*PRC2_X, lane 5—*Hs/Ac*PRC2_X_FL_EED (*Hs/Ac*PRC2_X with full-length EED), lane 6—Y641N *Hs/Ac*PRC2_X (protein degradation observed for Y641N PRC2_X is consistent with dynamic nature of this protein which leads to susceptibility to proteolysis) proteins used for crystallization and biochemical assays (5 μg per lane; previously frozen at −80 °C).

**Table 1 | Biochemical analysis of inhibitor 1 potency (*n* = 2 for each $K_i$; the data are mean ± s.d.).**

| Complex | $K_i$ (nM) |
|---|---|
| *Hs*PRC2_4 | 1.63 ± 0.13 |
| *Hs*PRC2_4 (Y641N) | 4.56 ± 0.16 |
| *Hs/Ac*PRC2_X | 3.10 ± 0.11 |
| *Hs*PRC2_4 (Y641N/Y661D) | 204 ± 16 |
| *Hs*PRC2_4 (Y111L) | 116.4 ± 0.7 |

Potency against *Hs/Ac* PRC2_X measured using rH3 substrate, intact *Hs*PRC2_4 and its primary oncogenic and secondary resistant mutants using oligonucleosome substrate.

against the N-terminal helix of the subunit, which is substantially less basic than in *Ac*EZH2.

SANT1 is connected to SANT2 via a structural motif spanning residues 260–308 that has a Cys$_3$His zinc-binding site. The corresponding structure is referred to as the motif connecting SANT1L and SANT2L (MCSS) in *Ct*EZH2. Although encompassing near-identical Cys$_3$His zinc-binding site, the rest of the MCSS structure is different in *Ct*EZH2 (Supplementary Fig. 3b) partly due to the sequence insertion in *Ct*EZH2. MCSS along with SANT2 provides for the most extensive contacts between EZH2 and the SUZ12-VEFS subunits (Fig. 4a). Another stretch of dynamic polypeptide chain connects MCSS to the SANT2 domain defined by residues 432–472. This linker is naturally missing in *Ct*EZH2.

Although juxtaposition of three helices in the two SANT domains is very similar (Fig. 4b), SANT1 is nearly 70 residues in length with much longer first helix followed by a long flexible loop connecting it to the second helix. SANT2 on the other hand is much more compact, 40 residues in length. Although annotated

as a SANT domain, SANT2 resembles a nucleic acid recognition motif with significant proportion of basic residues, an estimated pI of 9.6, and a zinc coordinating site (evident in the electron density map but not modelled due to poorly defined coordinating polypeptide around it), suggestive of a role in nucleic acid recognition (Fig. 4a). SANT2L of *Ct*EZH2 bears significant structural similarity to the SANT2. Finally, SANT2 is connected via an engineered flexible Gly linker, unresolved in the crystal structure, to the C-terminal catalytic domain. The catalytic domain consists of a pre-SET region containing two CXC motifs coordinating six zinc ions and the SET domain. SET domain is also the site of the inhibitor recognition (Fig. 3a,c). Expectedly, catalytic domains of *Ac*EZH2 and *Ct*EZH2 in general, and the SET domain in particular show similar overall structures with two key differences (Supplementary Fig. 3c).

The C-terminal end of the structure reported here, often referred to as the post-SET, is largely disordered. In *Ct*PRC2, the corresponding region is structured and forms the roof of the SAM-binding pocket. Post-SET region in every known structure of the cofactor-bound SET domain from other lysine methyltransferases is seen to make the ordered lid of the cofactor-binding site as in *Ct*PRC2. Conversely, this region is disordered in SET domains in the absence of the cofactor, as is the case for the PRC2 structure reported here. Hence it is likely that the cofactor recognition induces the post-SET structure. Second, PRC2 structure reported here has completely modelled I-SET region, while *Ct*PRC2 I-SET helix is unwound and disordered at its C-terminal end in both the basal and the stimulated states. This is also the region where there is an insertion of seven residues in *Ct*EZH2 sequence relative to *Ac*EZH2. This is significant because, based on the results described below, I-SET contributes to the binding site of the PRC2 inhibitors currently in clinical development.

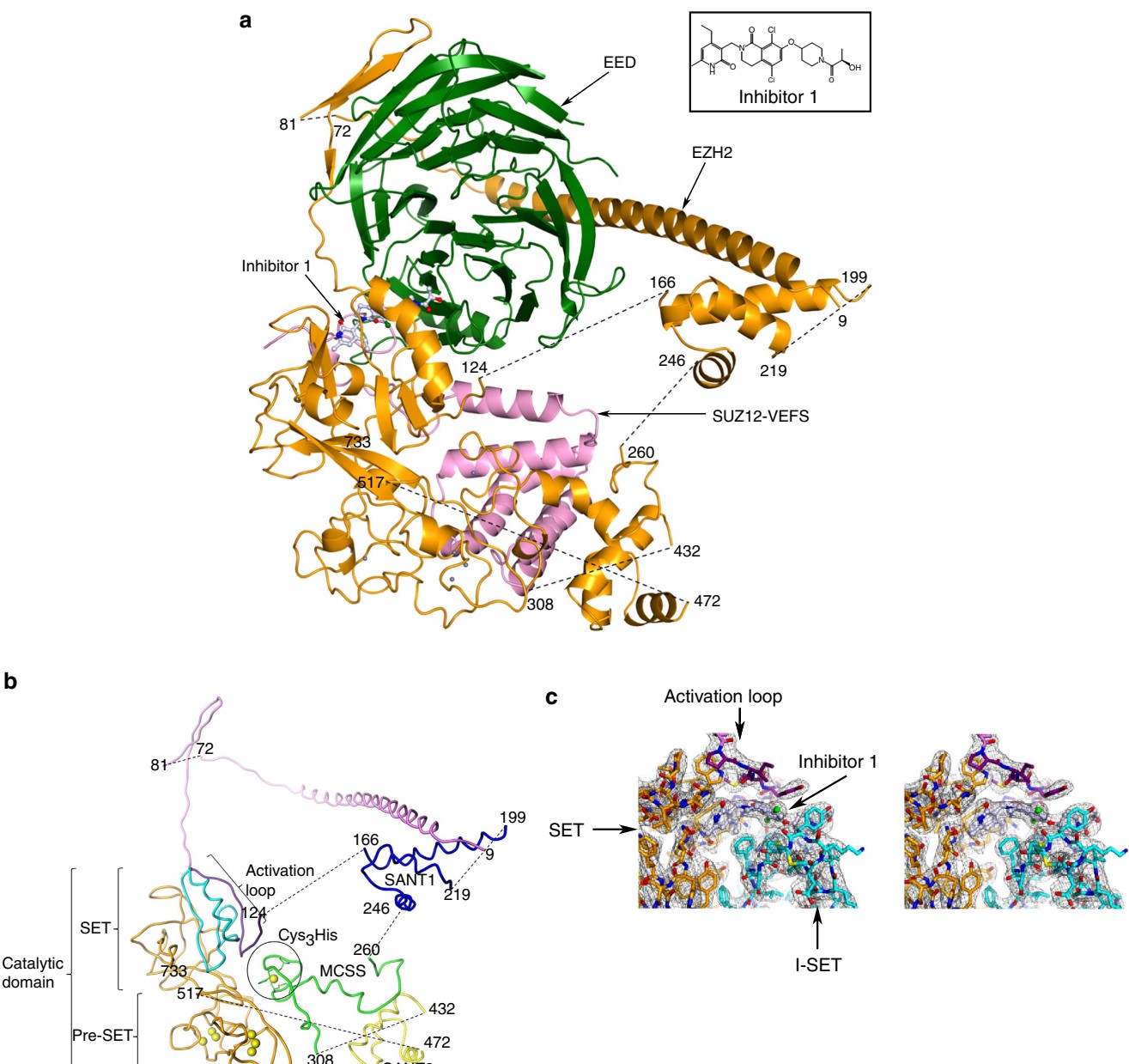

**Figure 3 | *Hs/Ac*PRC2 co-crystal structure.** (**a**) Complex structure with EZH2 (orange), EED (green) and SUZ12-VEFS (pink) in ribbon representation, and the inhibitor 1 in ball-and-stick. The dotted lines represent the disordered polypeptide chain of EZH2 not modelled in the structure. Zn atoms are shown as spheres. The inset shows the chemical structure of the inhibitor. (**b**) The subunit architecture of EZH2. Zn atoms are shown in CPK representation. The colour code is VIBGYO (of the rainbow colours, VIBGYOR) from the N- to the C-terminus. In addition, I-SET of the SET domain is highlighted in cyan. (**c**) Stereo drawing of 2Fo-Fc electron density contoured at 1σ around the ligand-binding site. The ligand is shown in ball-and-stick representation, while the protein atoms are shown as sticks. Colour code is the same as in **b**: SET domain is shown in orange, I-SET region of the SET domain in cyan and the N-terminal activation loop is shown in indigo colour.

SUZ12-VEFS structure begins with an N-terminal random coil of 25 residues (562–586) that is sandwiched between the EZH2 SET domain and EED, forming the glue that holds them together. The rest of the protein forms a helical bundle with exclusive interactions with different parts of EZH2 (Supplementary Fig. 4). The first four helices interact intimately with MCSS of EZH2 and the C-terminal two helices provide interaction surface to the two N-terminal helices of SANT2. *Ct*SUZ12-VEFS structure bears close resemblance to the *Hs*SUZ12-VEFS structure reported here.

EED structure is essentially identical to the previously determined crystal structures[4,33,34]. Its H3K27Me3 recognition site[4] is proximal to the EZH2 SET domain. *Ct*EED is a larger

protein (Supplementary Fig. 3a) with significant insertions and deletions relative to *Hs*EED, with structural differences in the corresponding regions. However, it provides similar interactions to partner proteins in the context of PRC2.

The overall architecture of the PRC2 *vis-à-vis* relative placement of the subunits in the structure described here is similar to that observed in *Ct*PRC2 (ref. 5; Supplementary Fig. 3).

**Role of EED and SUZ12 in the activation**. The structure of the isolated SET domain has been determined and is shown to be in the inactive state[9,10], consistent with the knowledge that EZH2 requires EED and SUZ12-VEFS partners to form a catalytically

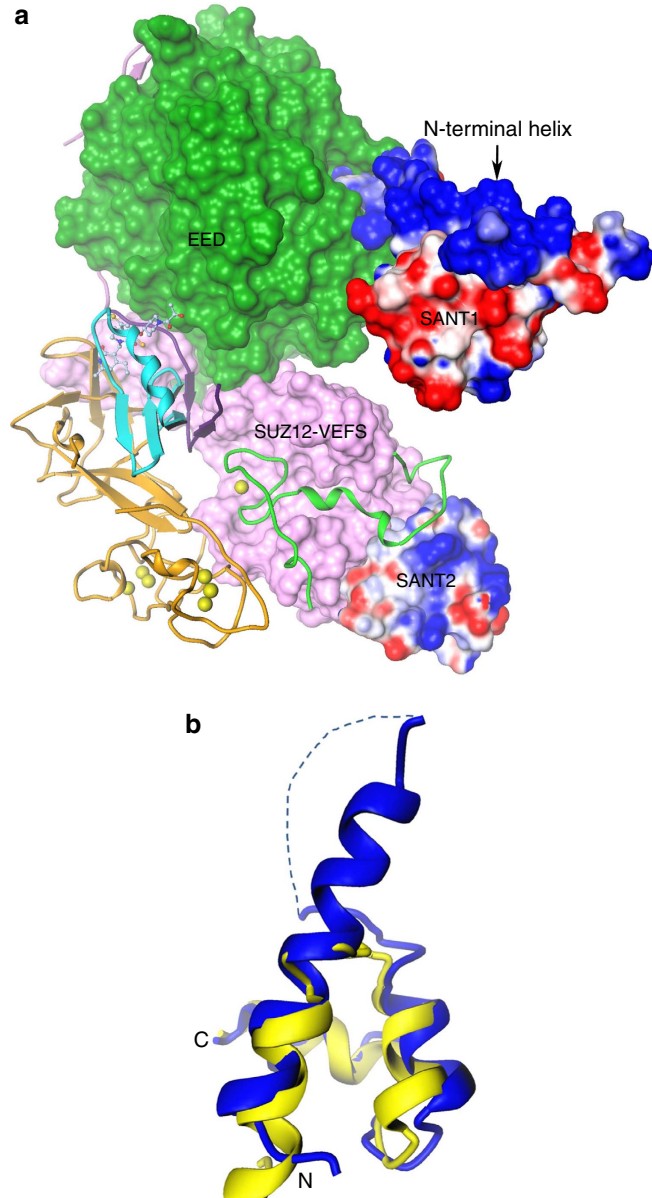

**a**

N-terminal helix

EED

SANT1

SUZ12-VEFS

SANT2

**b**

C

N

**Figure 4 | The SANT domains.** (**a**) *Hs/Ac*PRC2 structure is shown with EZH2 N-terminal helix and the SANT domains in electrostatic surface representation, and EED (green) and SUZ12 (pink) in Connolly surface representation. SANT1 is very acidic and packs against the basic N-terminal helix, while SANT2 is basic in nature. (**b**) Comparison of SANT1 domain (blue) with SANT2 (yellow) domain. Dotted line indicates the dynamic linker between the two N-terminal helices of SANT1. While overall architectures are very similar, SANT1 is larger than SANT2.

competent complex. The PRC2 structure suggests that EED and SUZ12-VEFS play the role of allosteric effectors in activating the SET domain. Activation loop occupies the critical position at the interface of the SET domain, EED and VEFS (Figs 3b and 4a, Supplementary Fig. 5). Modelling suggests that the structuring of the activation loop by EED may induce a conformational change in the I-SET region (residues 643–681; Fig. 5a), rendering cofactor as well as inhibitor recognition possible.

Activation loop of EZH2, located in the N-terminal half of the primary sequence, is brought into the proximity of the C-terminal SET domain largely through its interaction with EED. The proximal placement of the activation loop is not compatible with the known structure of the inactive SET domain (PDBID: 4MI5).

When the inactive SET domain is modelled into the PRC2, the activation loop is in severe steric conflict with I-SET of the inactive SET domain (Supplementary Fig. 5). We propose a model wherein the activation loop may play an important role in inducing a conformational switch of the SET domain. In this model, as EZH2 residues 81–124 wrap around EED, the activation loop is brought into the proximity of the SET domain. It may sterically induce I-SET into its observed conformation as exemplified by van der Waal's interaction between Tyr111 and Tyr661 (see below). This results in the rigid body twisting motion of the I-SET: the top of the I-SET moves away from the N-terminal segment, while the bottom (residues 643–645) moves closer to it and engages in reciprocal hydrogen-bonding interactions with the backbone atoms of residues 120–122, akin to a short parallel β-sheet (Fig. 5a, Supplementary Movies 1 and 2). The backbone-mediated specific hydrogen-bond interactions between the SET domain and the activation loop preclude the need for the sequence conservation in these regions, consistent with the observation of structural conservation but sequence divergence among other lysine methyltransferases (Fig. 5b).

VEFS may play a less prominent, but no less important, role in the ordering of the activation loop. The interpretation of the biochemical data on the patient-derived oncogenic mutations in SUZ12 in the context of PRC2 structure highlights the role of VEFS in PRC2 activation. The SUZ12-VEFS mutation W591C has been identified in T-cell acute lymphoblastic leukaemia patients[35,36]. The equivalent mutation in the *Drosophila* protein leads to severely compromised activity of the PRC2 (ref. 7). In the structure presented here, indole N of Trp591 is seen making a hydrogen-bonding interaction with the backbone carbonyl of Pro115, thereby contributing to the structure of the activation loop of EZH2 (Fig. 5c). The loss of the specific interaction accompanying W591C substitution may result in less-structured activation loop, which in turn manifests itself in compromised catalytic activity.

Another patient-derived loss-of-function mutation N618Y in VEFS[35] results in the failure of SUZ12 to assemble itself into PRC2 (ref. 7). *Hs/Ac*PRC2 structure shows that Asn618 is located buried in the VEFS—EZH2 interface making reciprocal interactions with the backbone peptide group of EZH2 Tyr292 (Fig. 5c). Substitution of Asn618 with a bulky aromatic Tyr at the severely constrained interface may result in compromising the integrity of the complex.

**Oncogenic Y641N PRC2.** In our attempts to structurally characterize Y641N *Hs/Ac*PRC2, we noticed that the solution behaviour of the mutant three- and five-protein PRC2 is significantly altered relative to the WT: the mutant is prone to dimerization (Supplementary Fig. 6). Monomeric Y641N PRC2 when analysed by HDX-MS shows increased baseline dynamics (of EZH2 subunit, Fig. 6a), suggestive of the significant and far-reaching consequences of the substitution.

Nonetheless, Y641N *Hs*PRC2_4 does remain sensitive to inhibitors currently in clinical development[21,23] and the inhibitor 1 reported here (Table 1), suggesting an overlap of the conformational space between the WT and the Y641N PRC2. This is further underscored by the crystal structure of the Y641N *Hs/Ac*PRC2 in complex with inhibitor 1 reported here. The mutant structure is nearly identical to the WT PRC2 described above (Fig. 6b).

**Inhibitor recognition and acquired resistance.** Pyridone series inhibitors of PRC2 activity under clinical development are extremely potent and selective, and have generally been anticipated to recognize the expected SAM-binding site based on the

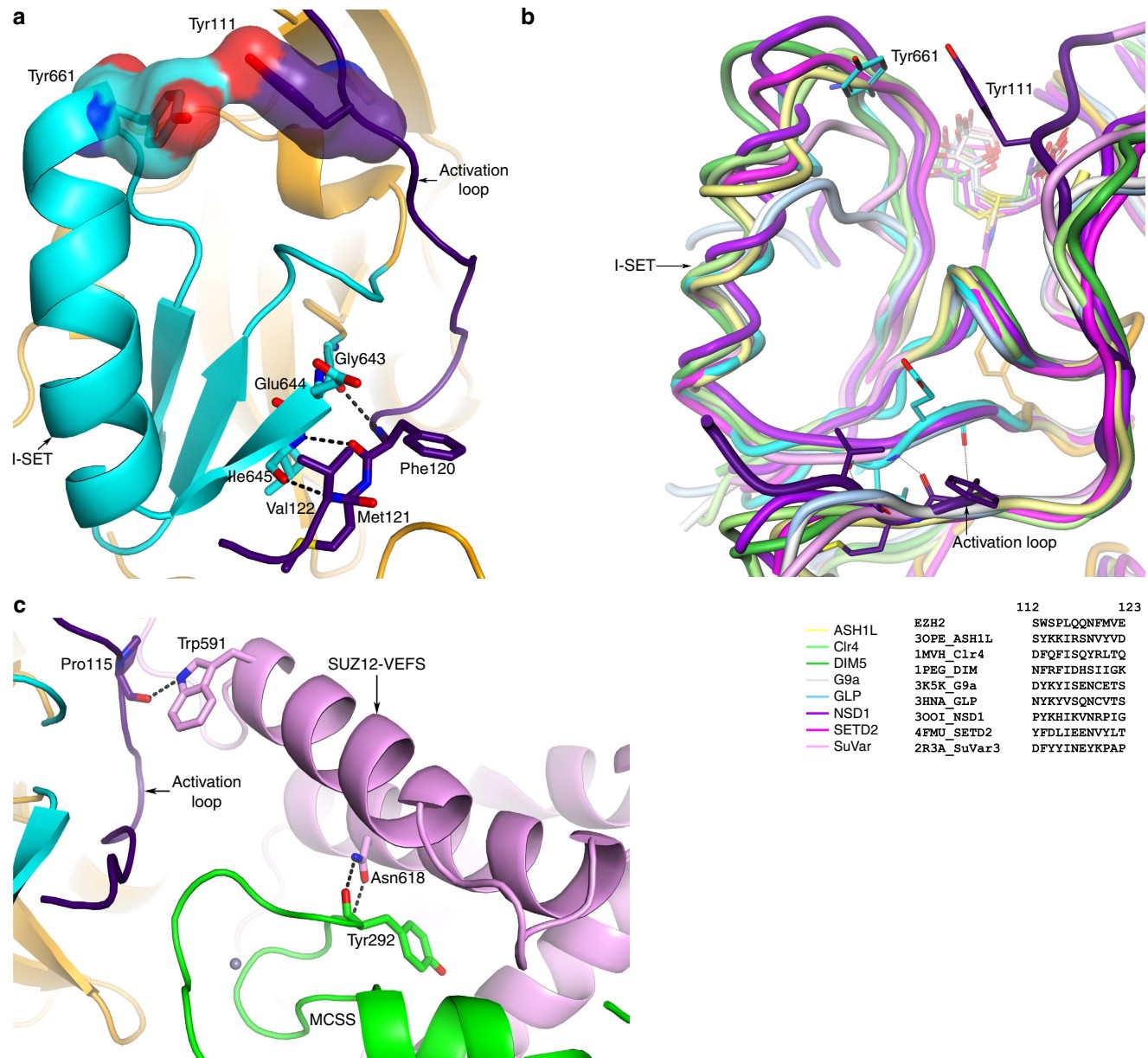

**Figure 5 | Stabilization of the active SET domain by the activation loop.** (**a**) Activation loop may induce the SET domain conformational change via specific backbone (bottom) and non-specific sidechain (top) interactions with I-SET. I-SET (Gly643–Gly681) is highlighted in cyan, the rest of the SET domain is shown in orange and the activation loop is shown in indigo. (**b**) Structural alignment of lysine methyltransferase SET domains with PRC2 SET domain shows structural conservation of the activation loop, with divergent sequences. Structure-based sequence alignment of this region shows poor overall sequence similarity. (**c**) Role of SUZ12-VEFS (shown in pink) in the stimulation of PRC2. Patient-derived SUZ12 mutations, W591C and N618Y abrogate favourable interactions, impacting the PRC2 activity/integrity. Colour code is same as in Fig. 3b.

modelling. However, interestingly, HDX-MS showed that the compound recognition not only lowers deuterium exchange around parts of the SET domain, but also significantly lowers deuterium uptake for the protein backbone around the residue 111 of the activation loop (Fig. 7a). The structure in complex with compound 1 reported here shows that the ligand recognition is nearly orthogonal to the SAM-binding mode (Fig. 7b). The pyridone group anchors the ligand through its reciprocal hydrogen-bonding interactions with the backbone of Trp624 from the conserved GXG motif of the SET domain (Fig. 7c). This region is also occupied by the homocysteine moiety of SAM for the specific recognition. Thus the partial overlap of the cofactor- and inhibitor-binding sites is consistent with SAM-competitive nature of these inhibitors (Fig. 7d). It is also consistent with previous

report that activation of PRC2 by H3K27Me3 mark results in the increased residence time and lower $K_i$ for certain PRC2 inhibitors exemplified by GSK126 but has no effect on cofactor and substrate binding, while impacting $k_{cat}$ of the activated enzyme[37]. It suggests that the inhibitor, while being SAM-competitive, has a distinct binding mode compared with SAM. Compound 1 has almost identical inhibition profile for both basal and stimulated states of *Hs*PRC2_4 (Supplementary Fig. 7).

Nonetheless, the rest of the ligand occupies a hitherto unanticipated pocket that is partially formed by the I-SET region from the SET domain as the base, and the activation loop forming the lid of the pocket. The oxygen atom of the carbonyl moiety of the lactam present in inhibitor 1 makes specific interaction with the backbone NH of Tyr111. Tyr111 from the activation

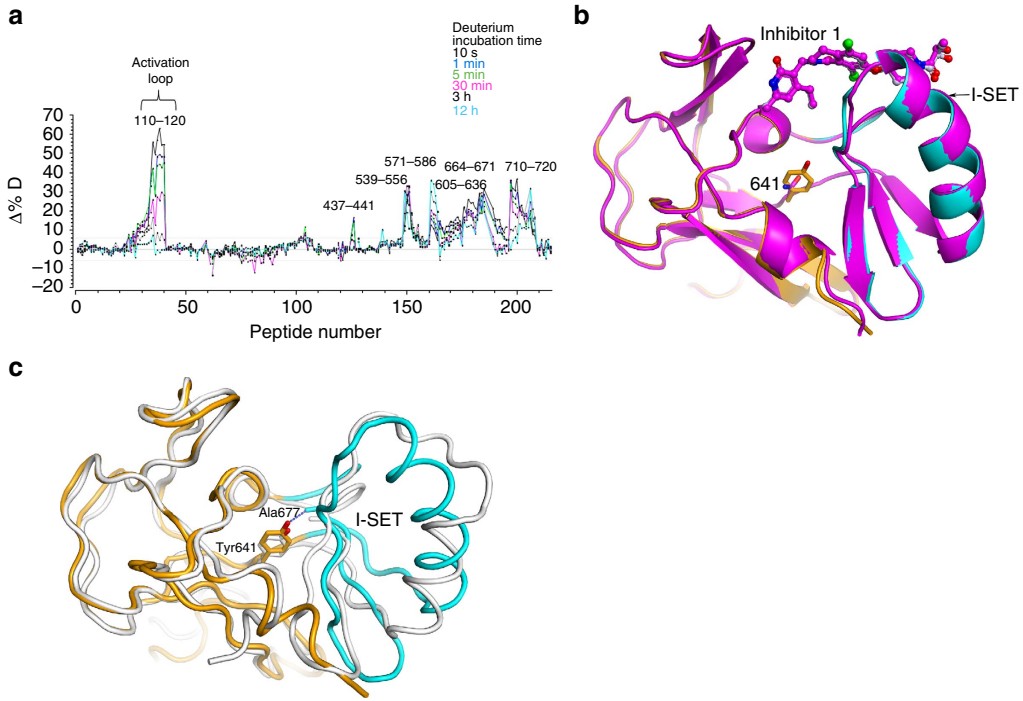

**Figure 6 | Y641N and the EZH2 dynamics. (a)** Differential (Y641N versus WT *Ac*EZH2 in the context of the crystallization constructs) deuterium uptake plots highlight increased dynamic nature of the C-terminal catalytic domain and the activation loop of Y641N *Ac*EZH2. Deuterium incubation times: 10 s, 1 min, 5 min, 30 min, 3 h and 12 h. **(b)** Inhibitor-bound structures of the WT (orange with I-SET highlighted in cyan) and Y641N (magenta) PRC2 are essentially identical. For clarity, only the SET domains bound to the inhibitor 1 are shown with the mutation site highlighted. **(c)** SET domain from PRC2 is shown in orange, with I-SET (Gly643–Gly681) highlighted in cyan. Isolated SET domain (PDBID: 4MI5) is shown in grey. Tyr641 and Ala677 are sites of primary oncogenic mutations, at 3.5 Å, in van der Waal's contact with each other.

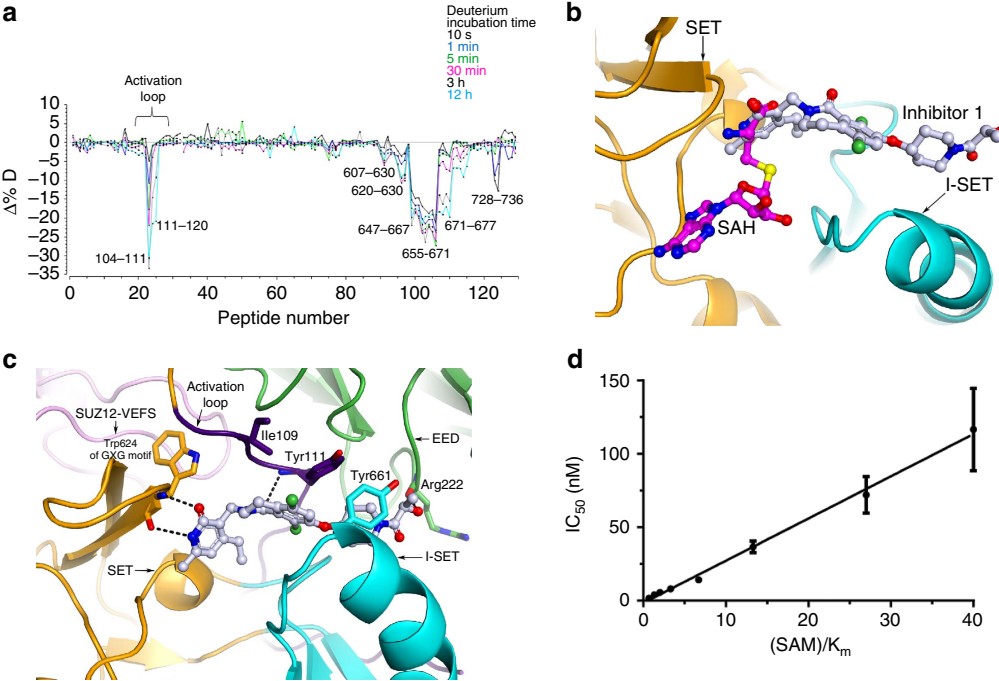

**Figure 7 | PRC2 recognition of inhibitor 1 is orthogonal to the SAM-binding mode and involves the activation loop. (a)** Differential deuterium uptake plots of EZH2 subunit of PRC2 with and without the inhibitor highlights dynamic changes in the activation loop, which is also the site of certain resistance-conferring mutations generated *in vitro* to first-generation PRC2 inhibitors. Deuterium incubation times: 10 s, 1 min, 5 min, 30 min, 3 h and 12 h. **(b)** Ligand binding relative to SAM recognition. SAM is modelled in purple based on the *Ct*PRC2 structure[5]. The SET domain is shown in orange with I-SET highlighted in cyan. **(c)** Some of the details of the inhibitor recognition are illustrated. Activation loop is shown in indigo and EED is shown in green. **(d)** Plot of $IC_{50}$ values of compound 1 as a function of SAM concentration relative to the $K_m$ of SAM ($(SAM)/K_m$) measured using histone methyltransferase activity assay with *Hs*PRC2_4 as described in the Methods section (plotted as mean ± s.d. of two independent experiments). These values show a linear relationship, as expected for a SAM-competitive inhibitor.

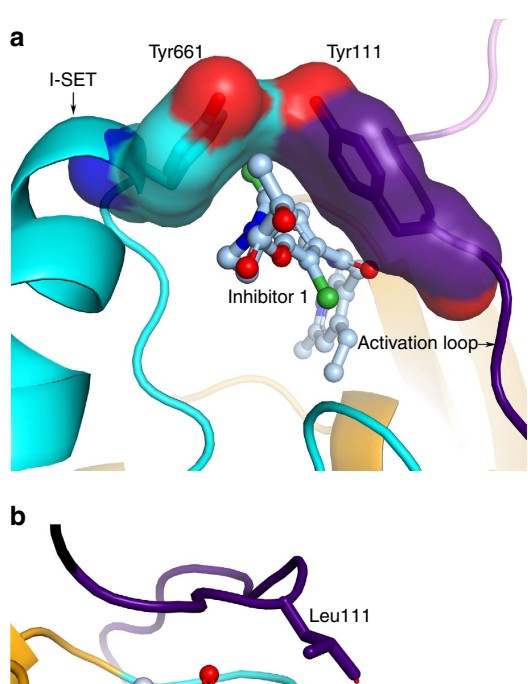

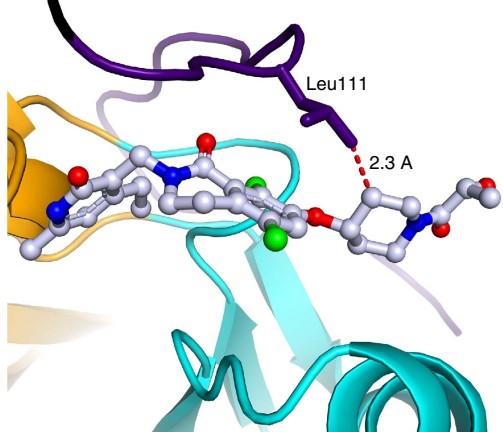

**Figure 8 | Inhibitor binding and the mechanism of drug resistance.**
(**a**) Ligand accesses the cavity enclosed by the contact between Tyr111 and Tyr661 in the active PRC2. When either of these two sidechains is mutated by a smaller one, the cavity would shrink in size posing a steric conflict for the ligand recognition. (**b**) Modelling of Y111L shows the introduction of the steric conflict with the ligand. Colour notation is same as in Fig. 3b.

loop and Tyr661 from the I-SET region are in contact with each other making a closed cavity that is accessed by the 'pendant' piperidine ring of the compound (Fig. 8a). The hydroxypropanoyl group emerging from the cavity makes a specific interaction with the backbone carbonyl oxygen of Arg222 from EED (Fig. 7c).

Given the anchoring role played by the pyridone, we believe that all known inhibitors of PRC2 with the pyridone group essentially have the same binding mode. Significant structural and sequence differences between the inhibitor-binding sites of CtEZH2 and AcEZH2 contribute to the divergence of the shape and the electrostatic nature of the inhibitor-binding pockets between the two PRC2 systems.

As promising as the pyridone inhibitors are for lymphoma patients, Karpas-422 and Pfeiffer cells, when grown under the selective pressure of inhibitors EI1 and EPZ-6438, acquire resistance to inhibition through specific mutations[38,39]. One site of mutation, Y661, is located in the I-SET. Another hotspot of mutations for acquired resistance maps to the activation loop; the specific mutations are I109K, Y111D, and Y111L. All three sites of acquired resistance mutation provide interactions for the molecular recognition of the inhibitor 1 (Fig. 7c). Therefore I109K, Y111D and Y661D could alter the ligand-binding potency by changing the electrostatic nature of the inhibitor-binding site. Single Y111L mutation induces ∼100–1,000-fold shifts in $IC_{50}$

values in one resistance model[39], while Y111D induces similar shift in another[38]. Y111L and Y641N/Y661D HsPRC2_4 also became refractory to biochemical inhibition by compound 1 (Table 1). We postulate that the loss of inhibitor potency due to mutations of Tyr111 and Tyr661 is intimately linked to the mechanism of PRC2 activation upon formation of the quaternary complex.

Pyridone inhibitors do not show any appreciable binding affinity for the isolated SET domain[9], presumably because the ligand-binding site exists only in the context of active SET domain conformation of PRC2. On the basis of the model proposed above, activation loop through Tyr111-mediated contact with the I-SET in general, and Tyr661 in particular creates a ligand-binding cavity that is accessed by the terminal 'pendant' group on the ligand (Fig. 8a). When one or both of the sidechains in contact (Tyr111 and Tyr661) are mutated to a smaller sidechain, the enclosed pocket formed by their contact in the catalytically competent state would shrink in size, resulting in the energetic penalty for the recognition of the 'pendant' group. Indeed, modelling suggests that Y111L introduces unfavourable steric conflict, impacting the compound 1 binding potency (Fig. 8b).

## Discussion

We have described the structure of PRC2 consisting of AcEZH2, HsEED and HsSUZ12-VEFS that could serve as a robust platform for structure-based drug design. Although similar in the overall architecture of the recently described CtPRC2 structure[5], sequence divergence and significant differences in the tertiary structure suggest that the system reported here may be a more accurate description of human PRC2, and a better model system for drug discovery.

Structural analysis of other SET domains suggests that the activation loop is a conserved structural feature of lysine methyltransferases, although it is divergent in the primary sequence (Fig. 5b). This conserved structural feature may play as important a role among other SET domain containing lysine methyltransferases. In PRC2, EED is seen to be playing an obvious role in structuring of the activation loop through its extensive interactions with EZH2 N-terminal segment. SUZ12-VEFS is seen to buttress the specific activation loop conformation through interactions with the backbone atoms of the N-terminal segment, thereby playing a role in the enzyme activation.

Comparison of the conformation of the SET domain in PRC2 with that of the isolated SET domain shows that on formation of the complex, the I-SET region (residues Gly643–Gly681) undergoes a rigid body rotation (Fig. 6c). Notably, Gly643 and Gly681 are invariant among SET domains of lysine methyltransferases; they may serve as the flexible 'hinge' about which the I-SET could 'swing' between its different conformational states. Conservation of the hinge Gly suggests that the conformational flexibility of I-SET may be an evolutionarily conserved negative regulatory control on the methyltransferase activity of the enzyme. Interestingly, two of the prominent primary oncogenic mutations identified in lymphoma patients[13,17], Y641N and A677G, are positioned in or around the I-SET (Fig. 6c). Tyr641 and Ala677 are highly conserved among lysine methyltransferases, are structural neighbours in van der Waal's contact with each other and form an axis around which the I-SET 'swings'. It suggests that substitution of either Tyr641 or Ala677 with a smaller sidechain may have qualitatively similar consequences, but proportional to the magnitude of change in the sidechain. In concordance with this hypothesis, although the WT PRC2 prefers unmethylated H3K27, A677G mutation leads to the loss of substrate specificity among unmethylated, mono- and

**Table 2 | Data collection, phasing and refinement statistics**

| | Native (wild type) | Anomalous | Native (Y641N) |
|---|---|---|---|
| *Data collection* | | | |
| Space group | P2$_1$ | P2$_1$ | P2$_1$ |
| Cell dimensions | | | |
| a, b, c (Å) | 69.6, 115.0, 153.0 | 72.7, 120.9, 149.8 | 71.3, 115.1, 153.94 |
| α, β, γ (°) | 90, 102.53, 90 | 90, 102.9, 90 | 90, 103.3, 90 |
| Resolution (Å) | 150–2.62 | 120–3.28 | 115–2.98 |
| R$_{merge}$ | 0.061 (0.306)* | 0.109 (0.658) | 0.099 (0.58) |
| I/σI | 16.6 (4.3) | 14.2 (2.9) | 13.0 (2.5) |
| Completeness (%) | 99.6 (99.9) | 97.7 (97.5) | 99.7 (99.6) |
| Redundancy | 3.4 (3.4) | 3.3 (3.2) | 3.4 (3.5) |
| | | | |
| *Refinement* | | | |
| Resolution (Å) | 150–2.62 | | 115–2.99 |
| Number of reflections | 70,801 | | 49,142 |
| R$_{work}$/R$_{free}$ | 0.191/0.238 | | 0.188/0.24 |
| No. of atoms | | | |
| Protein | 15,460 | | 15,467 |
| Ligand | 72 | | 72 |
| Zn | 14 | | 14 |
| Water | 36 | | 0 |
| B-factors | | | |
| Protein | 56 | | 59 |
| Ligand | 48 | | 45 |
| R.m.s.d. | | | |
| Bond lengths (Å) | 0.01 | | 0.01 |
| Bond angles (°) | 1.1 | | 1.16 |

R.m.s.d., root mean square deviation.
A single crystal was used for each data set.
*Highest resolution shell.

di-methylated H3K27 (ref. 17). Y641N PRC2, on the other hand with significantly larger change in the sidechain, overwhelmingly favours dimethylated H3K27 as the substrate[31]. The HDX-MS results suggesting greater dynamic character of the Y641N PRC2 (Fig. 6a) and the proclivity of the mutant to dimerize (Supplementary Fig. 6), further emphasizes the far-reaching structural consequences of this specific substitution.

Notably, acquired drug resistance models identify EZH2 mutations in regions that show poor sequence conservation among lysine methyltransferases and do not involve catalytically important residues. However, the region is important for the activation of the enzyme. Thus, analogous to kinase inhibitor terminology, pyridone inhibitors could be classified as type II PRC2 inhibitors that not only compete with the cofactor SAM, but also occupy the extended pocket that is unique to PRC2, accounting for their exquisite selectivity.

In conclusion, we have reported a crystal structure of three-component PRC2 that reveals the mechanism of EED and SUZ12-VEFS-mediated activation of the EZH2 enzymatic activity. It uncovers a surprising ligand-binding mode that reconciles published cellular data on drug resistance acquired through secondary mutations in apparently incongruous regions of EZH2. Importantly, the structure provides an unprecedented opportunity for design and development of the second-generation PRC2 inhibitors that could be less vulnerable to the development of acquired resistance, should this mechanism present itself clinically. In principle, these could be molecules that rely less on the interactions with the extended pocket for potency, and more on catalytically important residues—akin to type I kinase inhibitors. The structure also unveils a possibility to design selective covalent inhibitors of PRC2 activity through targeting of the poorly conserved Cys663 in the vicinity of the ligand-binding site. Such inhibitors could mean significant advance in epigenetic therapies for relevant cancer patient population.

## Methods

**Expression of PRC2 using multi-open reading frame vector.** To facilitate structural studies of PRC2, we have designed an expression vector which allows co-expression of PRC2 three-protein complex in insect cells. This vector is based on pFASTBacDual expression vector (Life Technologies). We have used internal ribosomal entry site (IRES) from the 5′-untranslated region of Perina nuda virus (PnV)[40] to drive the expression of both EED-PnV_IRES-SUZ12-VEFS polycitronic message under control of p10 promoter, while EZH2 gene and its variants were expressed under the control of polH promoter (Supplementary Fig. 8a). The FLAG purification tag was placed on SUZ12-VEFS subunit, which was assumed to be the subunit with the lowest expression level since its transcription is driven by IRES.

The protein constructs used in the structure determination included *Ac*EZH2 (G1KPH4; 1–736) with the internal deletion of amino acids 329–415 and replacement of residues 478–498 with the flexible (GGGGS) × 3 linker based on HDX-MS (Fig. 1), SUZ12-VEFS (545–726; S583D) and EED (81–441). All genes were codon optimized for mammalian expression and synthetized by GenScript. DNA encoding EED PnV IRES and SUZ12-VEFS was synthetized as single open reading frame and directly cloned into p10 promoter of pFASTBAC Dual (Life Technologies) using *Sma*I and *Xho*I. The resulting vector was used for subcloning of genes encoding EZH2 protein variants into polH promoter multiple cloning site of pFASTBAC Dual using *Bam*HI and *Not*I. Viruses were generated using standard Bac-to-Bac viral generation protocols (Life Technologies) and amplified to high-titer passage two (P2) stocks. Protein overexpression was conducted in exponentially growing Sf9 or Sf21 insect cells (depending on the optimum cell line for specific constructs) infected at 2 × 10$^6$ with P2 viral stock at MOI = 1. PRC_4 was generated by co-infection of EZH2, EED, RbAp48 and SUZ12 viruses at MOI = 1 in Sf9 insect cells. The sequence encoding FLAG purification tag was placed on EED subunit.

**Purification of PRC2.** All WT and crystallized 3-component PRC2 were purified identically. They were purified from cell lysate using Flag affinity chromatography. Cells were lyzed in 50 mM Tris 8.0, 200 mM NaCl, 5% glycerol, 0.25 mM TCEP supplemented with EDTA-free protease inhibitor cocktail (Roche). In all, 1.5 ml of lysis buffer was added per 1 g of frozen biomass. The clarified lysate was obtained by centrifugation of cell lysate at 10,000*g* for 1 h at 4 °C. A total of 5 ml of Anti-FLAG M2 Agarose (Sigma) was added per 5 l of biomass and incubated for 3 h at 4 °C (batch binding). Flag resin bound to PRC2 was washed with 20 column volumes (CV) of 50 mM Tris pH 8.0, 200 mM NaCl, 5% glycerol, 0.25 mM TCEP followed by elution of PRC2 using 3 CV of 50 mM Tris 8.0, 200 mM NaCl, 5% glycerol, 0.25 mM TCEP supplemented with 200 µg ml$^{-1}$ of FLAG Peptide (DYKDDDDK). As expected, single-step FLAG purification allowed stoichiometric

capture of all the three desired subunits of PRC2. PRC2 was further purified using S200 26/600 column (GE Healthcare) pre-equilibrated with 2 CV of 25 mM Tris pH 8.0, 200 mM NaCl, 5% glycerol, 0.5 mM TCEP buffer. Peak fractions containing PRC2 were concentrated to 10–20 mg ml$^{-1}$, flash-frozen in small aliquots using liquid nitrogen and stored at $-80\,^{\circ}$C (Fig. 2c, Supplementary Fig. 8b). PRC2_4 complex was purified using Flag affinity chromatography as described for the 3-component PRC2.

Y641N PRC2 mutant purification was conducted essentially as described above. Fractions containing Y641N PRC2 monomer were pooled together and concentrated to 7–9 mg ml$^{-1}$ while checking aggregation profile by aSEC (Supplementary Fig. 6). Y641N PRC2 retained its monomeric behaviour in this concentration range. Freshly purified Y641N PRC2 was used for crystallization and HDX-MS, since samples stored at $-80\,^{\circ}$C showed increased dimer fraction and failed to crystallize.

**Crystallization of PRC2.** Hs/AcPRC2 (10–25 mg ml$^{-1}$) was pre-incubated with inhibitor 1 at 1:5 molar ratio for 1 h at 4 °C before crystallization set up. The crystals of WT and Y641N Hs/AcPRC2 used for X-ray data collection grew in hanging drops with micro-seeding by mixing 1–2 μl PRC2-inhibitor complex and 1–2 μl of reservoir solution (0.1 M Bis-tris, pH 5.2–6.6 or 0.1 M MES, pH 5.2–6.4, 21–27% mePEG 2 K, 5 mM TCEP, pH 7.0) at 13 °C. Crystals were flash-frozen in liquid nitrogen after transferring to 2 μl reservoir solution containing 25% (v/v) glycerol as a cryoprotectant and then stored in liquid nitrogen.

**X-ray data collection and structure solution.** All data were collected at APS IMCA-CAT beamline 17-ID (SAD at 1.28293 Å wavelength and the native at 1 Å) at 98 K and processed using autoPROC[41]. There are two Hs/AcPRC2 units per asymmetric unit. Structure solution was obtained using MR-SAD. Partial model was obtained by molecular replacement using EED[33] (PDBID: 2QXV) followed by the SET domain[10] (PDBID: 4MI0) as the search models in Phaser[42]. Complete model was built using Zn-SAD experimental phases calculated using autoSHARP[43]. The structure was refined using 2.62 Å native data collected at 1 Å wavelength and the refinement program CNX[44]. Final rounds of refinement were conducted using the program autoBUSTER[45]. The twofold non-crystallographic symmetry averaging was used during refinement in CNX. The refined structure served as the starting model for the rigid body refinement of Y641N Hs/AcPRC2 structure in complex with the inhibitor 1. WT PRC2 structure has 88.6% of residues in the most favoured regions, 11.1% in the additional allowed regions, 0.2% in generously allowed regions and 0.1% in disallowed regions of the Ramachandran plot. Corresponding Ramachandran statistics for Y641N PRC2 are 88.1, 11.4, 0.4 and 0.1%, respectively.

**HDX mass spectrometry.** Five-protein PRC2 (EZH2, EED, SUZ12, RBBP4 and AEBP2 (209–503)) was diluted to 10 μM in working buffer (20 mM tris pH = 7.2, 50 mM NaCl, 2 mM TCEP) for HDX-MS analysis. The HDX-MS system is equipped with a specially configured Pal HTX-xt Autosampler (Leap Technologies), where deuterium exchange was initiated at 4 °C with addition of 40 μl of D$_2$O solution (20 mM Tris pH 7.2, 50 mM NaCl) to 4 μl of protein sample. Deuterium exchange was conducted across five time points (10 s, 60 s, 10 m, 1 h, 10 h) in duplicate. The exchange was arrested by the addition of 20 μl of cold quench buffer (4 M guanidine hydrochloride, 5 mM TCEP) to give a final pH ∼2.5. Samples were injected and digested in-line at 49 μl min$^{-1}$ across an immobilized protease XIII/pepsin column (w/w, 1:1; NovaBioAssays) maintained at 4 °C in the Leap cooler box. Peptides were collected and washed on a BEH C4 (2.1 mm × 5 mm) trap column (Waters), and subsequently eluted for separation across a BEH C4 (1 mm × 50 mm) analytical column using a gradient ramp of acetonitrile delivered by Ultimate 3000 nano pumps (Dionex). The Velos Pro OrbiTrap (Thermo Fisher Scientific) mass spectrometer electrospray source was held at 200 °C to minimize deuterium back-exchange. However, when initially generating peptide pools, the source was operated at 325 °C to promote peptide identification. A peptide list was initially generated with Proteome Discoverer (version 1.4, Thermo Fisher Scientific) with a 'no enzyme' search, and low-confidence peptides were removed (mass tolerance <5 p.p.m.). Peptide pools were then exported and measured for deuterium incorporation using HD Examiner 2.0 (Sierra Analytics) software. Low-confidence peptides for tracking deuterium incorporation were further removed, as well as shorter peptides of less than five residues in length. Denaturing conditions (quench time, quench composition, LC run time and gradient) were independently optimized to collectively provide the most comprehensive sequence coverage for all PRC2 proteins (EZH2—90%, SUZ12—98%, EED—87%, RBBP4—98% and AEBP2—89%). Differential plots, comparing deuterium uptake between two selected states, were generated in HD Examiner 2.0.

Comparative HDX data sets were collected within the same-day batch run to minimize potential fluctuations across separate data collection sets. In addition, blank injections were run between each sample to ensure low protein carryover <5%. Deuterium uptake was calculated in HDExaminer for each validated peptide, based on number of peptide residues (number of residues-(prolines, and two n-terminal amide hydrogens)). No full-exchange protein was used as a reference for relative uptake changes, and back-exchange was not corrected. Relative changes in uptake were reflective of the absolute difference in deuterium incorporation

percentage between two protein states (for example, WT versus mutant). A separate assessment of deuterium uptake variability from replicate runs of a control protein provided a significance threshold of ±6% difference in deuterium uptake for a single time point. This difference threshold is similar to those defined previously by other HDX-MS studies[46]. The ±6% deuterium threshold is illustrated in the residual plots generated by HDExaminer to compare the percentage incorporation change between two protein states. In this study, the primary regions of significant deuterium uptake difference were in excess of 20% difference across multiple time points (10 s, 1 m, 5 m, 30 m, 3 h and 12 h), well above the significance threshold.

**SAXS data collection and processing.** SAXS data were collected at the SIBYLS beamline at the Advanced Light Source in Berkeley CA. Solution samples were prepared by buffer exchange using dilution of protein sample in the desired buffer followed by re-concentration from frozen protein samples (15–20 mg ml$^{-1}$) purified as described above. SAXS measurements were conducted on PRC2 samples at 1, 3 and 10 mg ml$^{-1}$. Buffers with no protein were used for background subtraction from protein solutions. Data collection proceeded as previously reported[47,48]. In brief, samples were placed 1.5 m from a MAR165 CCD detector arranged coaxial with the monochromatic beam set at 1 Å; 10$^{12}$ photons s$^{-1}$ were impingent on the sample. Buffers were collected before and after samples. Buffer subtraction and raw image data were integrated by beamline software specific for this arrangement. Scattering data were plotted on log of x-ray intensity scale versus momentum transfer ($q$) in inverse Å, where $q = (4\pi \sin(\theta/2))/\lambda$, $\theta$ is the scattering angle relative to the incident beam, $\lambda$ is the wavelength. Subtraction of either buffer yielded identical results to within experimental error (∼1% of signal).

Processing of SAXS data was conducted utilizing the ScÅtter package available at www.bioisis.net. To optimize the signal to noise for each construct and remove artefacts, three concentrations of samples were collected at one-third dilutions from the highest concentration after purification. Highest concentrations ranged between 10 and 5 mg ml$^{-1}$. The resulting samples were exposed for 0.5, 0.5, 2 and 4 s for data collection. The two 0.5 s exposures for each concentration were referenced against one another to check for radiation damage and as none was observed, the two exposures were averaged. The longer exposures were used to reduce noise in the high q region. Once each individual concentration had been merged, the three concentrations were used to apply a concentration-dependent correction. Minor concentration dependence was observed. The ScÅtter package allows for the extraction of R$_g$, mass[27], volume[27] and Porod exponent[28] from scattering data. GNOM[29] was utilized to extract the P(r) function and further the maximum dimension of each construct.

**Radioactive filter-binding assay.** Oligonucleosomes were purified from HeLa cells as previously described[48]. Recombinant PRC2 was combined with methyl-acceptor substrate in 100 mM Tris pH 8.5, 0.01% Tween-20 and 4 mM DTT. In all, 44 μl was added to a microtiter plate containing 1 μl of 100% DMSO or test compound. SAM was prepared by combining ³H-labelled SAM with unlabelled SAM in assay buffer, such that the specific activity was 0.4 μCi μl$^{-1}$. The reaction was initiated by adding 5 μl of SAM to the microtiter plate. Following incubation at room temperature, the reaction was stopped with the addition of 100 μl of 20% TCA. The tritiated-methyl-acceptor product was captured using a 96-well filter plate (MSIPN4B50, Millipore) and washed five times with phosphate-buffered saline buffer. Scintillation fluid (50 μl) was added to the dried filter plate and counted in a liquid scintillation counter. In assays where several different constructs were compared, we first established linearity of ³H-labelled SAM incorporation over a 120-min period and then 30 min reactions using 7.5 nM of PRC2 and its variants with 3 μM of SAM and specific concentrations of various substrates, as described below, were performed in duplicates. The concentration of various methyl-acceptor substrates were as follows: oligonucleosomes purified in-house at 0.05 mg ml$^{-1}$, rH3.1 (NEB) at 0.28 μM, rH3K27Me2 (active motif) at 0.28 μM and H3 peptide (1–35; Anaspec) at 3 μM. The total counts for three independent experiments were plotted as mean ± standard deviation. For $K_i$ determinations, 5 nM PRC2 and 15–50 μM SAM were used and reactions were run for 30–60 min. $K_i$ values were determined by fitting background corrected counts to the Morrison equation for competitive tight binding inhibition using SAM $K_m$ values for each individual enzyme[49] (Prism, Graphad Software Inc.).

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

## Acknowledgements

We thank all members of Oncology Structural Biology and Protein Sciences Group for help, discussions and technical advice related to this manuscript. We thank Greg Hura (SAXS Consulting) for SAXS data analysis of PRC2 variants. SAXS data were collected at SIBYLS through its mail-in programme that is funded by DOE BER IDAT and NIH MINOS (RO1GM105404). We are also thankful to Martin Edwards for his support and attention to this work. We gratefully acknowledge Anne Mulichak, Kevin Battaile and Eric Zoellner for data collection at APS IMCA-CAT 17-ID, Wei Wang for obtaining $^{13}$C NMR spectrum of inhibitor 1, Mike Gehring for External Research Solutions support and members of Oncology Research Unit for encouragement.

## Author contributions

A.B. and W.L. engineered, cloned and purified proteins; K.S.G. performed structure determination; Y.-L.D. crystallized protein; B.B. performed HDX-MS experiments; P.B. and K.A.M. performed the biochemical assays; Y.-A.H., W.D. and X.Y. provided reagents; N.G. expressed protein; P.-P.K. and S.S. designed and synthesized inhibitor 1; A.B., K.S.G., K.A.M., X.Y. and A.E.S. designed research and analysed data. A.B. and K.S.G. prepared the manuscript.

## Additional information

**Accession codes:** Coordinates and associated structure factors have been deposited in the protein data bank with PDB ids 5IJ7 (*Hs/Ac*PRC2) and 5IJ8 (Y641N *Hs/Ac*PRC2).

**Competing financial interests:** All authors are employees of Pfizer, Inc.

**Reprints and permission** information is available online at http://npg.nature.com/ reprintsandpermissions/

