## [Peer Review File · Nature Communications]

Reviewer #1 (Remarks to the Author)

The revised manuscript is suitable for publication in Nature Communications in principle. The following list of minor issues concerning compound synthesis and characterization should be addressed before publication.

1. Synthesis of 296g: (a) MS of 296a: 335 maybe the peak of [M+Na] instead of [M+1]; (b) ¹HNMR of 296b: missing 1 H (amide NH); (c) ¹HNMR of 296b: missing 2 hydrogens; and (d) MS of 296e: [M+1] peak should be 246 instead of 245.

2. Synthesis of Cpd EE: (a) The alkyne starting material for the synthesis of Cpd AA is wrong. It should be "3-hexyn-2-one" instead of "pent-3-yn-2-one". This should be corrected in both the reaction scheme and experimental procedures sections; and (b) There is no characterization of any intermediates, from Cpd AA to Cpd DD.

3. Synthesis of 1: There is no characterization of intermediates 2, 3 and 4.

4. In addition, experimental procedures were not carefully written. There are a number of grammar and spelling mistakes that should be corrected.

Response to reviewer's comments Reviewer #1 (Remarks to the Author):

The revised manuscript is suitable for publication in Nature Communications in principle. The following list of minor issues concerning compound synthesis and characterization should be addressed before publication. 1. Synthesis of 296g: (a) MS of 296a: 335 maybe the peak of [M+Na] instead of [M+1]; (b) ¹HNMR of 296b: missing 1 H (amide NH); (c) ¹HNMR of 296b: missing 2 hydrogens; and (d) MS of 296e: [M+1] peak should be 246 instead of 245. Synthesis of 296g: (a) MS = 335 changed to [M+Na]; (b) The missing 1H in the ¹H NMR of 296b (amide NH) was added. Its chemical shift was 11.5 ppm as a broad singlet; (c) ¹H NMR of 296e from a different experiment had

all 9 protons present and accounted for; and (d) MS of 296e: [M+1] peak found to be 246, as expected. 2. Synthesis of Cpd EE: (a) The alkyne starting material for the synthesis of Cpd AA is wrong. It should be "3-hexyn-2-one" instead of "pent-3-yn-2-one". This should be corrected in both the reaction scheme and experimental procedures sections; and (b) There is no characterization of any intermediates, from Cpd AA to Cpd DD. Synthesis of Cpd EE: (a) The alkyne starting is indeed "3-hexyn-2-one". This was corrected in both the reaction scheme and experimental procedure; and (b) ¹H NMR and MS data were added for intermediates from Cpd AA to Cpd DD. 3. Synthesis of 1: There is no characterization of intermediates 2, 3 and 4. ¹H NMR and MS data were added for intermediates 2 and 3. We have MS data for intermediate 4 but no ¹H NMR data at this time and no intermediate 4 remaining to take the ¹H NMR. 4. In addition, experimental procedures were not carefully written. There are a number of grammar and spelling mistakes that should be corrected. The experimental procedures were corrected for grammar and spelling mistakes.